# Solitonic Windkessel Model for Intracranial Aneurysm

**DOI:** 10.3390/brainsci12081016

**Published:** 2022-07-31

**Authors:** Hiroshi Ujiie, Yoritaka Iwata

**Affiliations:** 1Department of Neurosurgery, Kamagaya General Hospital, Chiba 273-0121, Japan; huji-neuro@tbz.t-com.ne.jp; 2Ujiie Neurosurgical and Medical Clinic, Chiyoda-ku, Tokyo 102-0094, Japan; 3Faculty of Chemistry, Materials and Bioengineering, Kansai University, Osaka 564-8680, Japan

**Keywords:** Windkessel model, intracranial aneurysm, subarachnoid hemorrhage, soliton

## Abstract

The Windkessel model, which is known as a successful model for explaining the hemodynamic circulation, is a mathematical model with a direct correspondence with the electric circuit. We propose a theoretical model for the intracranial aneurysm based on the Windkessel-type steady blood flow. Intracranial aneurysms are well known vascular lesions, which cause subarachnoid hemorrhages. Since an aneurysm is an end-sack formed on the blood vessel, it functions as an unusual blood path that has characteristic features such as a reservoir and bottle neck orifice. We simulate an aneurysm by an electric circuit consisting of three different impedances, resistance, capacitance and inductance. A dumbbell-shaped aneurysm is the most dangerous aneurysm to easily rupture. Our aneurysmal model is created as a two-story aneurysm model for this point, thus namely the five-element Windkessel. Then, the mathematical formula was solved in numerical simulations by changing the size of the aneurysm and the elasticity of the aneurysm wall. An analysis of this model provided that the presence of the daughter aneurysm and the thinning of the aneurysm wall are positively correlated with a sharp increase in blood pressure in the aneurysm dome. Our mathematic aneurysm model proposes a good analogue to the real aneurysm and proved that this model includes soliton that is a non-decreasing wave propagation.

## 1. Introduction

The subarachnoid hemorrhage (SAH) is one of the main causes of sudden death, and it is triggered by the sudden rupture of intracranial aneurysms [1,2,3,4,5]. Therefore, SAH prevention has an important role for public health. The prediction of aneurysm rupture is still an on-going argument. The most important method to predict the aneurysmal rupture is diagnostic imaging such as size and shape [1,2,3,4,5]. This is a basic study to elucidate the mechanism of an aneurysm rupture related with its morphology. An intracranial aneurysm is a kind of blind end sack on the arterial wall that acts as a reservoir for circulating blood and the complex blood path within the end sack [6,7,8,9]. The blood flow into the aneurysm is influenced by the size of the aneurysm orifice and vessel geometry [6,7,8,9]. The inflow and the outflow path through the same orifice, and the flow collision of the inflow and outflow at the neck results in a slower vortex flow within the aneurysm. The aneurysm with the daughter aneurysm such as the bleb (so called doubly composed aneurysm) is reported to have a stagnant flow [4]. In such a bleb, the fragility of the aneurysmal wall is also observed. Recent advances in imaging proved that the fragility of the aneurysmal wall induced the deformation of blebs along the cardiac cycle [10]. The blebs must be the upper part of the two-story aneurysm that must be locally weak or fragile for inflammation [11,12].

Recently, CFD (computational fluid dynamics) is considered as the most popular tool to evaluate the hemodynamics affecting aneurysmal rupture There are many augments about low or high shear stress, which plays an important role in aneurysmal rupture. CFD for aneurysmal studies seem to have many limitations. In particular, CFD have not succeeded in predicting the rupture caused by the slow recirculation flows or stagnant flow conditions found in daughter aneurysms [13,14,15]. Maybe in such conditions, inflammation or mass transport between the blood and aneurysmal wall must be important [16]. On the other hand, the Windkessel model, which is well known as a successful model for explaining the hemodynamic circulation, is a mathematical model with a direct correspondence with the electric circuits [17,18]. The Windkessel model is a lumped model that is not suitable for the assessment of spatially distributed shear stress and aspects of flow travel; however, it is a simple and fairly accurate approximation of arterial afterloads. In order to explain the rupture of the doubly composed intracranial aneurysm, we construct a theoretical model for the intracranial aneurysm based on the Windkessel-type modeling.

In this paper, we propose a five-element Windkessel-type model for the two-story aneurysm and also identify the rupture condition of the intracranial aneurysm. We are the first to use the Windkessel model to evaluate aneurysmal flow, and are successful in demonstrating some resonating beats and the soliton wave propagation must be a key factor for aneurysmal rupture. 

## 2. Theoretical Model

### 2.1. The Five-Element Windkessel Model

The Windkessel model is known as a nonlinear steady flow model for explaining hemodynamic circulation. While the more detailed turbulence and vortex formation are not considered in Windkessel models, steady fluid mechanics are taken into account. Here, we propose a model for describing an intracranial aneurysm (Figure 1A), which will clarify a substantial role of steady fluid dynamics in the aneurysmal rupture.

To define the blood flow in the aneurysm based on the Windkessel-type modeling, it must be hypothesized that the size of the neck is the resistance of blood flow entering the aneurysm: *R* (resistance), the size of the aneurysm is the reservoir for blood: *C* (capacitance) and the inertial effect of aneurysmal wall: *L* (inductance). The resistance *R* is a typical resistance to block the blood flow into the aneurysm, because the blood flow entering the aneurysm increases if a larger neck is formed. The capacitance *C* is mainly determined by the size of aneurysm, thus blood flow into the aneurysm must be determined by aneurysm volume. Then, aneurysm shape and aneurysmal wall elasticity must affect the intra-aneurysmal flow; the elastic compliance effect is expected to appear if the elasticity of the aneurysmal wall increase. Therefore, an inertial term *L* is added to the Windkessel model.

The electrical analog of the two-story aneurysm model consisting of the mother and daughter aneurysm on the parent artery is shown in Figure 1B. In this model, the capacitor *C*_1_ and *C*_2_ represent the compliance of mother and daughter aneurysm, and the inductance of *L*_1_ and *L*_2_ stands for the inductance of the mother and daughter aneurysm, and *F, F*_1,_ and *F*_2_ are the blood flow in the parent artery, in the mother aneurysm and in the daughter aneurysm, respectively. According to Kirchhoff’s law, the following three equations are logically introduced. Let *t* be the time variable. The proposed model, which corresponds to the five-element Windkessel model (Figure 1), reads
(1)$$\begin{array}{ccc} R\left(I\left(t\right)-I_{1}\left(t\right)\right)=V\left(t\right) & & R\left(F\left(t\right)-F_{1}\left(t\right)\right)=P\left(t\right)\\ \frac{Q_{1}\left(t\right)}{C_{1}}+L_{1}\frac{dI_{1}\left(t\right)}{dt}=V\left(t\right) & \Leftrightarrow & \frac{Q_{1}\left(t\right)}{C_{1}}+L_{1}\frac{dF_{1}\left(t\right)}{dt}=P\left(t\right)\\ \frac{Q_{2}\left(t\right)}{C_{2}}+L_{2}\frac{dI_{2}\left(t\right)}{dt}=\frac{Q_{1}\left(t\right)}{C_{1}} & & \frac{Q_{2}\left(t\right)}{C_{2}}+L_{2}\frac{dF_{2}\left(t\right)}{dt}=\frac{Q_{1}\left(t\right)}{C_{1}} \end{array}$$
where *F*(*t*) and *P*(*t*) mean the blood flow and the blood pressure of the parent artery, respectively. Indeed, for the two models in Equation (1), electric currents *I*(*t*), *I*_1_(*t*), *I*_2_(*t*) and electric voltage *V*(*t*) are replaced with blood flows *F*(*t*), *F*_1_(*t*), *F*_2_(*t*) and blood pressure *P*(*t*), respectively. The quantity *Q*_1_(*t*) and *Q*_2_(*t*) denote the amount of blood inside the mother and daughter aneurysms, respectively. More precisely, *Q*_1_(*t*) and *Q*_2_(*t*) means the amount of blood inside the mother and daughter aneurysms, respectively. The *R* is the resistances associated with the neck size, which is taken to be very high for most of the blood flows going through the parent vessel. The flows *F*_1_(*t*) and *F*_2_(*t*) can be either positive, zero or negative (for the default direction of the flows inside the aneurysms, see Figure 1A). Consequently, the intracranial aneurysm is mathematically modeled in the form of the Windkessel model.

Let local pressures *P*_1_ and *P*_2_ correspond to the pressures on the capacitor *C*_1_ and *C*_2_, respectively. Much attention is paid to the effect due to the existence of the daughter aneurysm. Here, blood pressure *P*(*t*) and the blood flow *F*(*t*) in the parent artery are assumed to be given. In this situation, the flows *F*_1_(*t*) and *F*_2_(*t*) inside the aneurysms are the unknowns, as well as the local pressures *P*_1_(*t*) and *P*_2_(*t*) inside the aneurysms, so that the model equation is equivalent to solve
(2)$$\frac{d^{2}}{dt^{2}}F_{2}\left(t\right)+\frac{1}{C_{2}L_{2}}F_{2}\left(t\right)=\frac{d}{dt}\left[\frac{P\left(t\right)}{L_{2}}-\frac{L_{1}}{L_{2}}\frac{d}{dt}\left(F\left(t\right)-\frac{P\left(t\right)}{R}\right)\right]$$
in terms of *F*_2_ (*t*). This equation is obtained by beginning with differentiating the third equation of Equation (1), and next by substituting the second and first equations into it. This is the second-order differential equation, where the right-hand side of this equation is given by *P*(*t*) and *F*(*t*). Note that, in the present situation of simulating a blood flow inside human brain, *P*(*t*) and *F*(*t*) are given. By giving initial values *F*_2_(0) and *dF*_2_(0)/*dt*, Equation (3) leading to *F*_2_(*t*) is solvable. In this paper, *F*_2_(0) and *dF*_2_(0)/*dt* are fixed to 0.58 mL/min and 0 mL, respectively. On the other hand, the first equation of Equation (1) leads to *F*_1_(*t*). Once *F*_1_(*t*) and *F*_2_(*t*) are obtained, *Q*_1_(*t*) is calculated by the second equation of Equation (1), and then *Q*_2_(*t*) is obtained by third equations of Equation (1). *P*_1_(*t*) and *P*_2_(*t*) are obtained by *Q*_i_(*t*) = *C P*_i_ (*t*) (cf. *Q* = *CV* for electric circuits).

In terms of making a large-scale systematic calculation, the differential operators are discretized by the finite difference method. This treatment is rather legitimate in the present setting with the given *P*(*t*) and *F*(*t*). Indeed, a relatively large amplitude effect represented by the right-hand side of Equation (2), which plays a role of external force in the equations of motion, prevents the accumulation of errors.

### 2.2. Solitonic Modeling in Some Specific Cases

Another mechanism included in the present Windkessel-type steady flow is the soliton propagation. The solitons are the stably-traveling waves without changing shape and velocity [19,20,21,22,23]. Generally speaking, soliton propagations can be found in many scales and in many mediums [24,25,26]. According to exactly the same setting as Equation (1), we have
(3)$$P_{2}-P_{1}=L_{2}\frac{dF_{2}}{dt},\;P_{1}-P_{0}=L_{1}\frac{dF_{1}}{dt},\;\frac{dQ_{1}}{dt}=F_{1}-F_{2},$$
where we introduce additional quantity: local pressure *P*_0_ corresponding to the pressure on the resistance *R*. Note that the local pressure *P*_0_ is temporally introduced to simply discussion in this section, and it is not necessary in the main analysis. Equation (3) leads to
(4)$$P_{2}+P_{0}-2P_{1}=-L_{1}\frac{d^{2}Q_{1}}{dt^{2}}+\left(L_{2}-L_{1}\right)\frac{dF_{2}}{dt}.$$

The second term of the right-hand side can be negligible if *L*_1_∼*L*_2_ or *dF*_2_/*dt*∼0 is satisfied. Since the values of *L_i_* is associated with the elastic compliance, the cases with *L*_1_ < *L*_2_ tend to be realized at the initiation stage of the aneurysm formation. Let the capacity of the mother aneurysm be represented by the inverse dependence on the blood pressure *P*:*C*_1_ = −(*aP* + *b*) ^−1^ with constants *a* and *b*. By definition, the volume of the mother aneurysm is calculated as
(5)$$\begin{array}{c} Q_{1}=\int_{0}^{P_{1}}C_{1}dP=-\int_{0}^{P_{1}}{\left(aP+b\right)}^{-1}dP\\ =-\underset{\hat{p}\to 0}{\lim}\frac{1}{a}{\left[\log\left(p+\frac{b}{a}\right)\right]}_{\hat{p}}^{P_{1}}=-\underset{\hat{p}\to 0}{\lim}(\frac{1}{a}(\log\left(P_{1}+\frac{b}{a}\right)-\log\left(\hat{p}+\frac{b}{a}\right)))\\ =\frac{-\underset{\hat{p}\to 0}{\lim}1}{a}(\log(\frac{P_{1}}{\hat{p}+\frac{b}{a}}+\frac{b}{a}\frac{1}{\hat{p}+\frac{b}{a}}))=\frac{-1}{a}(\log\left(\frac{a}{b}P_{1}+1\right)) \end{array}$$
where $\hat{p}$ is an intermediate constant satisfying $\hat{p}+b/a=b/a$. As a result, let *L*_1_∼*L*_2_ or *dF*_2_/*dt*∼0 be satisfied, and
(6)$$\frac{L_{1}}{a}\frac{d^{2}}{dt^{2}}\log\left(\frac{a}{b}P_{1}+1\right)=2P_{1}-P_{2}-P_{0}$$
is obtained. This is the soliton equation admitting 1-soliton solution. In particular, this equation holds the same form as the Toda lattice equation [27,28,29,30,31], so that the local pressure *P*_1_ inside the mother aneurysm possibly holds the 1-soliton solution. Consequently, the local pressure in the mother aneurysm is in the non-decaying solitonic state, if the elastic compliance of mother and daughter aneurysms are not so different. As this model holds the soliton-type, non-decaying wave propagation, we call the present model the solitonic Windkessel model. In the case of soliton appearance, the solitonic wave propagation might contribute directly or indirectly to the rupture of the daughter aneurysm.

## 3. Method

Systematic calculations, including almost 30 different settings, are carried out for Equation (2). Equation (2) is discretized by the finite difference method with its incrimination unit *dt* = 0.002 sec. The blood pressure *P*(*t*) and the blood flow *F*(*t*) in the artery are given by
*F(t*) = 10 sin(2π*ft*) + *F*(0),(7)
*P*(*t*) = 20 sin(2π*ft*) + *P*(0),
where *f*, *F*(0) and *P*(0) are the frequency of the heartbeat, initial blood flow and initial blood pressure, respectively. For the human being, the typical blood flow of the middle cerebral artery is around 50 mL/min, the typical blood velocity is 60 cm/sec, the blood pressure is between 80 to 120 mmHg and the heart rate is from 40 to 120 beat per minutes with mean value 70 (Table 1), so that if we take the initial blood flow *F*(0) = 50 mL/min, the initial blood pressure *P*(0) = 100 mmHg, and *f* = 70/60 [1/sec] in the systemic calculations (Figure 2). In the natural situation, the blood flow in the parent artery (*F* + *F*_1_) is not so different before and after aneurysmal formation. To realize this feature, the amount of artery blood flow is adjusted by choosing the value of *R*, and the resistance parameter *R* is fixed to *R* = 2.2 for all of the systematic calculations.

## 4. Results

The blood flow and blood pressure before aneurysmal formation exactly corresponds to the natural situation shown in Figure 2. In other words, the blood flow and pressure shown in Figure 2 correspond to those pumping from the heart. Beginning with this natural situation, we simulate the change of blood flow and local blood pressure depending on the size and the elastic compliance of aneurysm.

Let us begin with a typical case of doubly composed intracranial aneurysms, in which a half-sized daughter aneurysm is formed on the mother aneurysm; the half size is realized by taking *C*_2_/*C*_1_ = 0.50, and the elastic compliance arising from the elasticity of aneurysmal wall are taken to be *L*_2_/*L*_1_ = 0.50. For a given setting: *R* = 2.2, *C*_1_ = 0.10, *C*_2_ = 0.05 and *L*_1_ = 10 and *L*_2_ = 20; Figure 3 shows the intra-aneurysmal flow and the blood flow of the parent artery. Less than 10% of the blood in the parent artery flows into the aneurysm. The blood flow inside daughter aneurysm includes an oscillation with a large frequency. Indeed, the composite oscillation formed in the daughter aneurysm holds a three times longer beat than the heartbeat. It implies the appearance of the resonating beat. Here, it is notable that, based on the ultrasonic measurement, the flow velocity of the ruptured aneurysm is observed to be relatively slow compared to those in the parent artery [9]. That is, distinct from the naïve expectation, a slower flow often leads to the rupture. As observed in the following, this fact is explained in the present model.

Figure 4 compares the oscillation of blood pressures for some different sizes of the daughter aneurysm. It simulates the growth of the daughter aneurysm from 1/5th the size to the same size of the mother aneurysm. The growth of the daughter aneurysm in an ordinary scenario, keeping *L*_2_/*L*_1_ = 2.0, is shown in the Figure 4A, and that in a soliton-appearance scenario, in keeping with *L*_2_/*L*_1_ = 1.0, is shown in the Figure 4B. A large value of the elastic compliance *L_i_* is expected to be due to the swelling of the aneurysm arising from the thinness of the blood wall. At the initial stage of the daughter aneurysm formation (*C*_2_/*C*_1_ = 0.20), the local blood pressures *P*_1_ and *P*_2_ satisfies |*P*_1_| > |*P*_2_|. However, at the latter stage with a sufficiently grown daughter aneurysm, the opposite relation |*P*_1_| < |*P*_2_| holds. Accounting for the situation with the similar sizes between mother and daughter aneurysms at the latter stages (*C*_2_/*C*_1_~1.0), a significant increase was detected only in the local pressure *P*_2_, where the local pressure *P*_1_ is kept to be less than 170 mmHg. The local blood pressure *P*_1_ of the mother aneurysms oscillates by the heartbeat cycle, while the local blood pressure *P*_2_ in the daughter aneurysm oscillates by both the heartbeat and another long beat. This fact coincides with the beating. Consequently, we found the appearance of the wave propagation entailing the resonating beats in the daughter aneurysm, where the resonating beat arises from the interference of flows with several different frequencies. Furthermore, independent of the compliance and of the scenario of aneurysmal evolution, we see that the local blood pressure *P*_2_ increases, as it is proportional to the size of the daughter aneurysm.

Figure 5 compares the oscillation of blood pressures for some different elastic compliances of the aneurysm, where only the mother aneurysm is formed. It simulates the evolution of the elastic compliance of aneurysm. Indeed, the elastic compliance should be different depending on elasticity and/or thinness of the aneurysmal wall. In Figure 5, the size of the aneurysm is fixed (*C*_1_ = 0.10), and only the compliance *L*_1_ is changed. The result shows us that even if the size of aneurysm is not well developed, very high local pressure of about 200 mmHg can be achieved by the compliance effect. Consequently, we see that the local blood pressure *P*_1_ increases, as it is proportional to the elastic compliance.

## 5. Discussion

### 5.1. Rupture Caused by the Size of Aneurysm

Let us begin with defining the aspect ratio *α*. The aspect ratio is defined by *α* = *d*/*w*, where *d* is the depth, and *w* is the neck width (Figure 1A). According to the surgical experience and the related studies, the risk factor of the intracranial aneurysm rupture is widely believed as the size larger than 10 mm [32] with or without the presence of the daughter aneurysm (e.g., with or without the entailing deformed shape). In a sense, the formation of the larger aneurysm is quantitatively identified by the aspect ratio larger than 1.6 [4,5]. In the previous studies [4,5], we reported that the aspect ratio of the unruptured aneurysm more than 1.6 is a border to start the pathological process to rupture.

The present five-element Windkessel model shows that the aneurysm with *C*_2_/*C*_1_ more than 0.8 is the dangerous state of the aneurysm (Figure 4), because intra-aneurysmal pressure becomes larger than 120 mmHg in all the cases, where 120 mmHg is the given maximum blood pressure from the heart. Let mother and daughter aneurysms be almost spherical, in which the radius of mother and daughter aneurysms are *R*_1_ and *R*_2_, respectively. In this situation, the following relations are approximately valid: (8)$$\frac{4\pi R_{1}^{3}}{3}=C_{1},\;\frac{4\pi R_{2}^{3}}{3}=C_{2},$$

It leads to the approximation of the aspect ratio
(9)$$\alpha =\frac{d}{w}\frac{\frac{3}{2}R_{1}+\frac{3}{2}R_{2}}{\frac{\sqrt{3}}{2}R_{1}\times 2}=\frac{\sqrt{3}}{2}R_{1}+\frac{\sqrt{3}}{2}{\left(\frac{C_{2}}{C_{1}}\right)}^{1/3},$$
where *w* and *d* are approximated by $\sqrt{3}$*R*_1_ and (2*R*_1_ + 3*R*_2_)/2, respectively (see Figure 1A). In this formalism, the aspect ratio of the dumbbell-shaped situation *C*_1_ = *C*_2_ is given by *α* = 1.73. Since the border of the rupture is identified to be *C*_2_/*C*_1_ = 0.8 based on the simulation, it leads to
(10)$$\alpha =\frac{\sqrt{3}}{2}+\frac{\sqrt{3}}{2}{\left(0.8\right)}^{1/3}=1.67.$$

It agrees well with the value *α* = 1.6 obtained by the experimental observation [4,5]. This finding proved the validity of the present five-element Windkessel model, and therefore is a substantial role of the static fluid dynamics effect.

### 5.2. Rupture Caused by the Formation of Daughter Aneurysm

The size ratio (more precisely, the volume ratio) between the mother and daughter aneurysm *Q*_2_/*Q*_1_ is essentially governed by the ratio between the proportional coefficients *C*_2_/*C*_1_ (cf. *Q* = *CV*). If the local blood pressure is sufficiently large enough to be larger than 180 mmHg, the rupture should take place. The possibility of rupture due to the daughter aneurysm is noticed; if the size (capacitance) of the daughter aneurysm becomes larger than 80% (*C*_2_/*C*_1_ = 0.80) of that of mother aneurysm (Figure 4), the rupture can take place. It tells us that the intra-aneurysmal pressure is demonstrated to be highly dependent on the size ratio *C*_2_/*C*_1_ between the mother and daughter aneurysm. The dumbbell shape formation, which has been believed to be a risk factor of the aneurysmal rupture, is confirmed quantitatively as the risk factor. Since the blood flow inside the daughter aneurysm cannot be as large in the present choice of the *R* value, the supersonic measurement for the rupture of the daughter aneurysm by slow blood flow is explained in the present model (Figure 6). Moreover, according to the present model, even if the size of the mother and daughter aneurysm (the values of *C*_1_ and *C*_2_) is not so large, the rupture can take place if the size ratio *C*_2_/*C*_1_ is large enough to be close to 1.0. Importantly, in such dumbbell shape formations, only the local pressure of the daughter aneurysm increases to possibly more than 200 mmHg.

### 5.3. Rupture Caused by the Compliance of Aneurysm

The compliance (*L*_1_ and/or *L*_2_) of the aneurysm arises from the elasticity or thinness of the aneurysmal wall. In Figure 5, the calculations performed by assuming a negligibly small daughter aneurysm are shown. The intra-aneurysm pressure increases linearly as the intra-aneurysmal compliance becomes larger. The fluctuation of the pulsatile pressure becomes extremely large, and is even comparable to some local pressures of the dumbbell-shaped aneurysms (see also Figure 4). The possibility of rupture due to the compliance of the aneurysm is noticed, and the intra-aneurysmal pressure is shown to be highly dependent on the elastic compliance *L*_1_ of the aneurysm. In this case, a rupture with a relatively small aneurysm can take place.

Recent studies of the motion of the aneurysmal wall on the basis of the dynamic digital subtraction angiography have demonstrated that the bleb of an aneurysm has a larger deformation amplitude than the aneurysmal sac [12,33,34]. In ruptured aneurysms, an area of pulsation was found to occur at the thinnest part of the aneurysmal wall, where it is covered with a fibrin plug [15,16]. The present model simply explains those previous clinical reports.

Previous reports have demonstrated that the main risk factor of the intracranial saccular aneurysm rupture is considered a size larger than 10 mm [32]. However, many ruptured aneurysms are small [3,35]. There is the important suspicion that many of the aneurysms may rupture within a short period before growing to a 10 mm size; the rest of the unruptured aneurysms stayed stable, and then may be found incidentally. We observed that the incidentally found aneurysm ruptured truly little, and most of the unruptured aneurysm stayed unruptured [36,37]. Thus, the risk of rupture of the aneurysms must be the highest at its early stage, because at the initiation, the aneurysmal wall must be very fragile.

### 5.4. Rupture Caused by Soliton Propagation

The soliton propagation is involved in our five-element mathematical Windkessel model. This is the first report to demonstrate soliton in the hemodynamic circulation. In particular, the local pressure *P*_2_ inside the daughter aneurysm holds the soliton solution. Comparing the Figure 4A and the Figure 4B, the soliton propagation contributes to increase the local pressure *P*_2_, even though the smaller *L*_2_ value is assumed in the solitonic situation (Figure 4B) compared to the ordinary situation (Figure 4A). Consequently, the local pressure in the daughter aneurysm is often governed by the non-decaying solitonic state, especially in cases when the aneurysm wall is not thin. The solitonic wave propagation contributes directly or indirectly to the rupture of aneurysm.

### 5.5. Rupture Caused by the Large Amplitude “to-and-fro” Oscillation

In the present model, the conditions such as the formation of the daughter aneurysm (*C*_2_/*C*_1_ > 0.80) and the appearance of aneurysmal compliance (*L*_1_ > 40) is directly associated with increased intra-aneurysmal pressure. It is estimated that, at the stage of aneurysmal initiation, the very thin aneurysmal wall grows rapidly (resulting in the larger *L*_1_ value and sometimes in the larger *C*_2_ value). As a result of resonating beats in both blood flow and local blood pressure, the thin aneurysmal wall is exposed by large amplitude wall movement (bulge and dents = to-and-fro movement along the heartbeat or another beating frequency). The to-and-fro movement of the fragile part quickly progresses into a vicious process to rupture (Figure 6). Without any elevation of the blood pressure, a rather small increase in intra-aneurysmal pressure induces to-and-fro oscillation for the thin and fragile aneurysmal wall. This must happen on real human SAH. 

## 6. Conclusions

The solitonic Windkessel model is introduced to explain the rupture of the intracranial aneurysm. A larger capacitance chamber in an electric circuit simulates a larger aneurysm. We further add inductance on the capacitance to express the compliance of the aneurysm. The linear dependence of the daughter aneurysmal pressure on the capacitance *C*_2_/*C*_1_ is noticed, and the linear dependence of the mother aneurysmal pressure on compliance *L*_1_ is noticed. The experimental observation of ruptured and unruptured aneurysms clarifies those difference that are observed in both aneurysmal shape and the hemodynamics. This fact agrees with the theoretical prediction based on CFD, in which the rupture is expected to take place, especially in aneurysmal parts with lower wall shear stress. Since the lower wall sheer stress is mostly realized by aneurysmal deformation (shape change), the shape of the aneurysm has been believed to be a primal factor of aneurysmal rupture. However, the shapes of ruptured and unruptured aneurysms are reported to be often quite similar without any significant differences [38]. The present model (solitonic Windkessel model) clarifies the other significant effects, such as the soliton propagation and the resonating beats. It simply tells us that the Windkessel-type modeling can take into account the effects essentially associated with the hemodynamic circulation, which are not easy to be incorporated in the fluid dynamic modeling.

Based on the five-element solitonic Windkessel model, we quantitatively explain the pathological process of aneurysmal rupture for the first time. The discovery is summarized as follows:The formation of a large aneurysm with an aspect ratio larger than 1.6;The formation of the daughter aneurysm as large as the mother aneurysm;The formation of a spherical-shaped aneurysm with large compliance.

These are quantitatively confirmed to be the risk factors of aneurysmal rupture. In particular, the present model succeeds in explaining the observed fact that a significantly slow flow inside the aneurysm and/or the formation of a small aneurysm can result in a rupture. It is worth noting here that, as is well known in the context of the usual application of the Windkessel model, the compliance of the blood vessel tends to be larger for younger patients and for female patients. The present quantitative conclusion, as for compliance, supports the fact that aneurysmal rupture often happens to young people. In terms of the compliance of the aneurysm:It is a non-decaying soliton propagation arising from small compliance;There is a “to-and-fro” large amplitude resonating oscillation arising from large compliance.

These are possible risk factors of aneurysmal rupture depending on the individual case. The first factor should be called the solitonic Windkessel effect, and the second factor should be called the ordinary Windkessel effect, that is highly affected by the resonating beat.

Consequently, for the utility, we clarify five typical properties of an aneurysm being close to rupture:(i)Aneurysm with an aspect ratio larger than 1.6 (cf. Section 5.1);(ii)Aneurysm with a thin or elastic blood wall, where the elastic situation sometimes happens to young patients (cf. Section 5.3);(iii)Dumbbell-shaped aneurysm with its daughter size as large as 80% of the mother aneurysm (cf. Section 5.2);(iv)Dumbbell aneurysms, especially the daughter aneurysm, with a thick or inelastic blood wall, possibly experience the soliton propagation. It explains that the formation of the daughter aneurysm itself is dangerous (cf. Section 5.4)(v)Dumbbell-shaped aneurysms with thin or elastic blood wall possibly experience the to-and-fro large amplitude resonating oscillation. This explains that the rapture can take place by even a small daughter aneurysm formation (cf. Section 5.5).

In the solitonic Windkessel model, ideal incompressible fluids with negligible viscosity are dealt with. The blood is a non-Newtonian fluid in more realistic setting, and it is interesting to see whether the viscosity and the other fluid dynamic features play a certain role or not.

## Figures and Tables

**Figure 1 brainsci-12-01016-f001:**
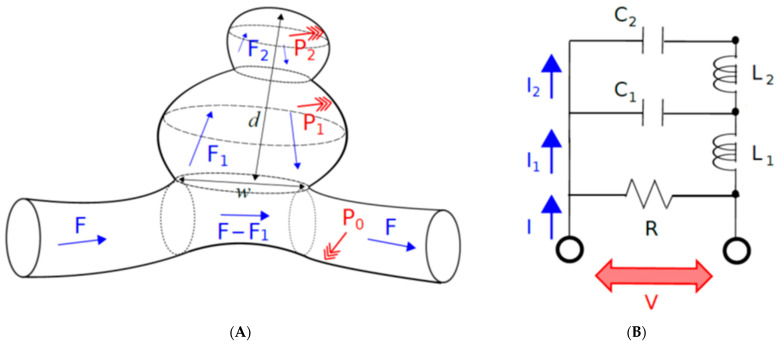
(**A**) Doubly composed intracranial aneurysm consisting of mother and daughter aneurysms (also called dumbbell-shaped aneurysm in the preceding works). The blood flow *F* is the amount of blood flow pumping from the heart, *F*_1_ is the blood flow in the mother aneurysm and *F*_2_ is the blood flow in the daughter aneurysm, where the circulative flows with the opposite direction are assumed in the aneurysms. (**B**) Solitonic Windkessel model for the doubly composed intracranial aneurysm. The corresponding electric circuit is shown, where *C*_1_ and *C*_2_ corresponds to the size/dimension of mother and daughter aneurysms, respectively. *R* determines the branching ratio *F*_1_ = *F* of the blood flow. The constants *L*_1_ and *L*_2_ are related to the elasticity of blood vessel wall. *P*_0_, *P*_1_ and *P*_2_ denote the local pressures, and *F*, *F*_1_, and *F*_2_ are blood flows.

**Figure 2 brainsci-12-01016-f002:**
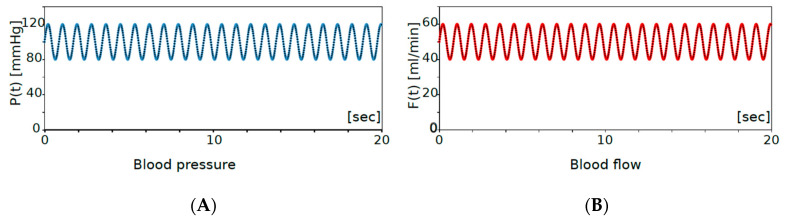
Blood flow *F*(*t*) and blood pressure *P*(*t*) in the main vessel are given in the left panel (**A**) and the right panel (**B**), respectively. Environmental variables are determined by the heartbeat: blood pressure *P*(*t*) and blood flow *F*(*t*) are plotted for 0–20 s. These are regularly provided by the heart. The time periodic oscillation is fixed by the typical beat 70 bpm (Table 1), the blood pressure is assumed to range from 80 to 120 mmHg and the blood flow to be 40 to 60 mL/min.

**Figure 3 brainsci-12-01016-f003:**
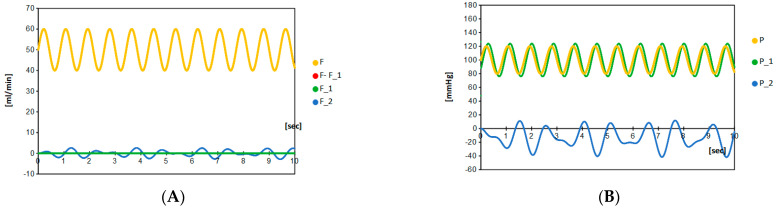
(**A**) The branching of blood flow from *F*(*t*) to *F*_1_(*t*) and *F*_2_(*t*) at a given setting (*R* = 2.2, *C*_1_ = 0.100, *C*_2_ = 0.005 and *L*_1_ = 10, *L*_2_ = 20). Here, we take *R* = 2.2 for the local blood flow in the main vessel *F*–*F*_1_ not to change too much (less than 10% in all the calculated cases) compared to *F*. It simulates the typical case of doubly composed aneurysm. Calculation shows that the blood flow in the artery *F* is almost equal to the blood flow in the parent artery *F*–*F*_1_, and therefore the blood flows in the aneurysms are kept to be small. Depending on the choice of four parameters (*C*_1_, *C*_2_, *L*_1_, and *L*_2_), flows *F*_1_ and *F*_2_ can be either positive, zero or negative. (**B**) In the same parameter setting, the local pressures *P*_1_(*t*) and *P*_2_(*t*) are calculated, which note that the blood pressure at the parent artery is equal to *P*(*t*) regardless of the choice of parameter. In this parameter setting, the amplitude of local pressure *P*_1_(*t*) is almost equal to *P*(*t*), while another local pressure *P*_2_(*t*) oscillates with irregular beat and small amplitude.

**Figure 4 brainsci-12-01016-f004:**
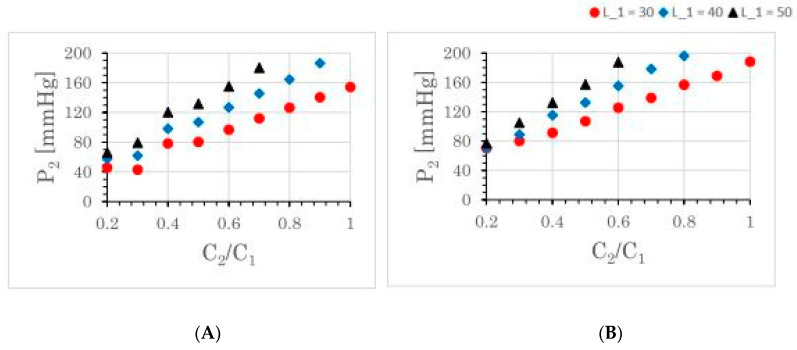
Maximum value of local blood pressure *P*_2_ depending on the size of daughter aneurysms. By fixing the size parameter *C*_1_ of mother aneurysm, the size ratio between mother and daughter aneurysms is changed as *C*_2_/*C*_1_ = 0.20, 0.30, … 1.00 for a fixed *C*_1_ = 0.10. *C*_2_/*C*_1_ = 1.00 corresponds to the ideal dumbbell shape. Note that the blood flow and blood pressure in the artery are almost unchanged (yellow and red ones in Figure 3), while local flows *F*_1_ and *F*_2_ and local pressures *P*_1_ and *P*_2_ easily change their maximum amplitude depending on the parameter. For the growth of daughter aneurysm, an ordinary scenario with *L*_1_ = 2*L*_2_ is demonstrated in the left panel (**A**), and a solitonic scenario with *L*_1_ = *L*_2_ is demonstrated in the right panel (**B**) (cf. Section 2.2). In all the presented cases, *P*_1_ is no more than 170 mmHg.

**Figure 5 brainsci-12-01016-f005:**
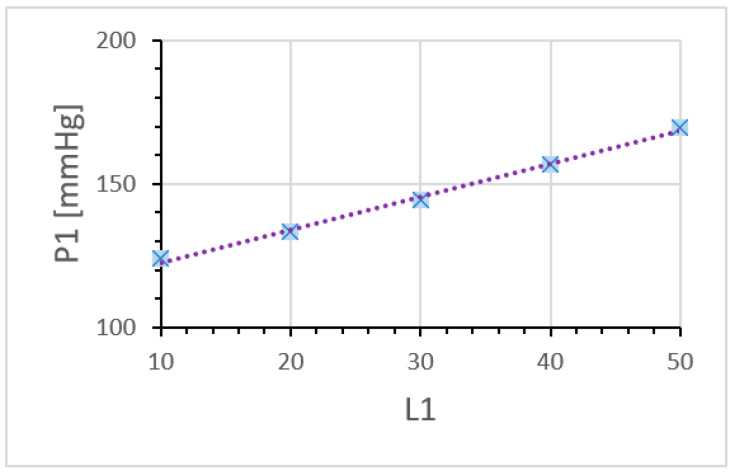
Maximum value of the local blood pressure *P*_1_ depending on the aneurysmal compliance *L*_1_. The daughter aneurysms are not assumed to be formed well (*C*_2_ = 0.1, *C*_2_ = 0.001, *L*_2_ = 0). By changing the compliance parameter *L*_1_ of mother aneurysm, the local pressure *P*_1_ of mother aneurysm increases linearly. Note that the blood flow *F*–*F*_1_ and blood pressure *P* in the parent artery is nothing different from those given by Figure 3.

**Figure 6 brainsci-12-01016-f006:**
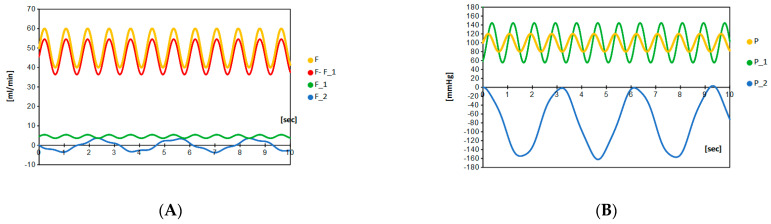
Blood flow (**A**) and blood pressure (**B**) at a given setting (*R* = 2.2, *C*_1_ = 0.100, *C*_2_ = 0.100 and *L*_1_ = 30, *L*_2_ = 60). Although the amplitude of resonating pressure |*P*_2_| becomes large, the resonating flow *F*_2_ is kept to be small enough to be less than 10 mL/min.

**Table 1 brainsci-12-01016-t001:** Parameters in realistic situations, which are actually adopted in the present calculation. As a result, the frequency *f* in this paper is fixed to 70/60 (1/sec), and the artery blood flow *F* to 50 (mL/min).

	Beat [bpm]	Artery Blood Flow [mL/min]
Typical values	70 (40–110)	50 (40–60)

## Data Availability

Not applicable.

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
