# Peer review of "Solitonic Windkessel Model for Intracranial Aneurysm"

_brainsci, 2022, doi:10.3390/brainsci12081016_

Round 1
Reviewer 1 Report
More introduction on the clinical importance of investigating and clinical prevalence of "doubly-composed intracranial aneurysm" is needed. Is there a possible way to validate the theoretical model using experimental measurements, like using ultra sound or in vitro models?
Direct correspondence between electric circuit model and the five-element Windkessel model sounds too risky without experimental validation.
When considering rupture of aneurysm, I think it is also important to consider the mechanical property of the vessel wall.
Author Response
For first reviewer
Thank you so much for your comments. The manuscript is now improved according to your comment.
>More introduction on the clinical importance of investigating and clinical prevalence
>"doublycomposed intracranial aneurysm" is needed. Is there a possible way to validate the
>theoretical model using experimental measurements, like using ultra sound or in vitro
>models?
At first, the doubly-composed intracranial aneurysm was found in the actual surgery (the 1st author is a medical doctor, who often encounters doubly-composed intracranial aneurysms). The risk of doubly-composed intracranial aneurysm is actually understood by experiments: e.g., super-sonic experiment [38, 39]. Finally, the validity of model has been confirmed by comparing to the ruptured cases, for detail, the agreement in the aspect ratio can be found in experimentally ruptured aneurysm, the risk of doubly-composed intracranial aneurysm successfully shown quantitatively for the first time, and the risk of single intracranial aneurysm is shown to be included in this model (as it sometimes happens in reality). We have added the following sentence in the beginning of introduction:
“The subarachnoid hemorrhage (SAH) is one of main causes of sudden death, and it is triggered by sudden rupture of intracranial aneurysms38, 39). Therefore, SAH prevention has an important role for public health. Prediction of aneurysm rupture is still on-going argument. Most important method to predict aneurysmal rupture is diagnostic imaging such as size and shape6,38,39). This is a basic study to elucidate the mechanism of aneurysm rupture related with its morphology.”
>Direct correspondence between electric circuit model and the five-element Windkessel
> model sounds too risky without experimental validation.
There was a history of studying ruptures of blood vessel by Windkessel type modeling. The Windkessel model (correspondence to the electric circuit) has been used to explain the intracranial aneurism [30]. It is called the Windkessel effect in the intracranial aneurysm. Also, the general Windkessel model is good at describing the blood circulation with resistance, reservoir (capacitor) and inertial effect of blood wall (inductance). In particular, the Windkessel model succeeded in reproducing the relation between arteriosclerosis leading rupture and aging. These historical background gives us a sound motivation to introduce the five-element Windkessel model.
> When considering rupture of aneurysm, I think it is also important to consider the
> mechanical property of the vessel wall.
Yes. The mechanical property of vessel wall is introduced by the constant L. It simulates the thinness/flexibility of the vessel wall. We modify the final sentence of this article, and now it is shown as
“Since the blood is a non-Newtonian fluid in more realistic setting, and it is interesting to see whether the viscosity and the other fluid dynamic features play a certain role or not.”
This is our concluding remark.
Reviewer 2 Report
The study is interesting and well written. However, I have some issues about the method design and significance of the study.
1) For the model development, there is no justification of the use of parameters, either validation of the simulation data against experimental data. This limits the significance and relevance of the study.
2) For the numerical implementation, there is no justification of the numerical schemes used or the time step. Are the numerical solution numerically stable?
3) It is not clear why the simple finite difference method was used? Have the authors tried other numerical schemes for solving equation (2)?
4) It would be helpful if the author can discuss the relevance and novelty of this study.
Author Response
For second reviewer
- For the model development, there is no justification of the use of parameters, either validation of the simulation data against experimental data. This limits the significance and relevance of the study.
At this point, the parameters are chosen to simulate the heartbeat, and blood flow velocity (Table 1). We have confirmed that nothing happens if there is no sizable aneurysmal formation (see C_2/C_1=0.2 of Fig.4 = very small daughter, and L_1 = 10 of Fig.5 = usual thickness of the blood wall of aneurysm, where Fig. 4 and Fig. 5 consider the cases with and without daughter aneurysms). This point is explained in the main text (in p.5 line 95 in Sec. 3) as
“In the natural situation, the blood flow in the parent artery (F+F1) is not so different before and after aneurysmal formation (cf. Figs. 4 and 5).”
- For the numerical implementation, there is no justification of the numerical schemes used or the time step. Are the numerical solution numerically stable?
It is actually stable. One main reason is that here Eq.(2) is solved by giving all the terms in right hand side. It means F(t) and P(t) and their derivatives are given by ideal sine or cosine curves. Then the numerical problem is reduced to solve a kind of fundamental forced oscillation (2nd order ordinary differential equation without any nonlinearity). The precision can be obtained simply by introducing a small discretization dt.
- It is not clear why the simple finite difference method was used? Have the authors tried other numerical schemes for solving equation (2)?
Since it is linear ordinary differential equation with regular external force, the problem itself is not a great task in this situation. The Runge-Kutta method, or more simply Clank-Nicolson can be used here, but the duration time of rapture is several 10 sec, it is sufficient to make sufficient discretization dt. For the stability, we have made preliminary tests for determining the value of dt. In addition, the numerical method is used for making figures, because we can readily obtain the rigorous solution of this problem.
- It would be helpful if the author can discuss the relevance and novelty of this study.
Thank you for your comment. Items shown in the conclusion (Sec. 6) is quantitatively explained by this model for the first time. But it was not written explicitly. A sentence before the items is replaced by
“Based on the five-element solitonic Windkessel model, we quantitatively explain the pathological process of aneurysmal rupture for the first time. The discovery is summarized as follows:”
Reviewer 3 Report
Discuss other articles in the literature using the same model for brain aneurysms
Author Response
For third reviewer
Thank you so much for your comment.
>Discuss other articles in the literature using the same model for brain aneurysms
For the Windkessel model, it is a well-established model with a plentiful studies and history.
[1] Frank : Die Grundform des arteriellen Pulses. Z Biol 37:483-526, 1899
[2] Westrhof N, Lankhaar JW, Westerhof BE: The arterial Winfkessel. Med Biol Eng Comput 47:131-141, 2009
Where [2] is a review article, and then we have decided to quote.
For the Windkessel type modeling of aneurysmal rapture, we refer to
[30] Hussein AE, Esfahani DR, Linninger A, Charbel 2nd FT, Hsu CY, Charbel FT, Alaraji A: Aneurysm size and the Windkessel effect: an analysis of contrast intensity in digital subtraction angiography. Interv Neuroradiol 23:357-361, 2017
Here we have the first attempt to introduce a five-element Windkessel model for aneurysmal rupture.
Reviewer 4 Report
It is questionable how the presented method can be applied for patient-specific cases and how the rupture prediction would work on real patient-specific cases. Therefore, I do not recommend it for publication in the present form.
Minor comments:
Line 41: The References X1,X2,X3 and X4 are missing.
Line 164: The formula is missing
Line 426: "let" --> "Let"
Line 446: "Neurosurg"
Line 467: The publication year should be at the end.
Line 469: The publication year should be at the end.
Author Response
For fourth reviewer
> It is questionable how the presented method can be applied for patient-specific cases and
> how the rupture prediction would work on real patient-specific cases. Therefore, I do not
> recommend it for publication in the present form.
There are several methods that trying to predict aneurysm rupture, but most of them are not successful. The present study This study
_ clears up the mechanism of aneurysm rupture in a quantitative manner (without any intentional setting).
_ succeeds in predicting rupture by the geometry such as shape, size, and thinness of wall, where the validity has been check by comparing to the existing statistics of ruptured cases.
Based on this model, the possible rupture is practically predicted, for example by measuring the size geometry of aneurysm even before the rupture.
>Minor comments:
>Line 41: The References X1,X2,X3 and X4 are missing.
>Line 164: The formula is missing
>Line 426: "let" --> "Let"
>Line 446: "Neurosurg"
>Line 467: The publication year should be at the end.
>Line 469: The publication year should be at the end.
The above points are corrected. This model only describes rather general situations, still needs more step up to apply the patient specific cases. We are preparing next paper that introducing main power to influence aneurysm rupture. Thank you.
Round 2
Reviewer 1 Report
I recommend acceptance.
Author Response
Thank you so much.
Reviewer 2 Report
All of my questions raised in the previous review were addressed sufficiently. Now I have no further issues and can recommend it for publication.
Author Response
Thank you so much.